# Increased Functional Connectivity Involving the Parahippocampal Gyrus in Patients with Schizophrenia during Theory of Mind Processing: A Psychophysiological Interaction Study

**DOI:** 10.3390/brainsci13040692

**Published:** 2023-04-20

**Authors:** Xucong Qin, Huan Huang, Ying Liu, Fanfan Zheng, Yuan Zhou, Huiling Wang

**Affiliations:** 1Department of Psychiatry, Renmin Hospital of Wuhan University, Wuhan 430060, China; psy_qxc@163.com (X.Q.); lexie_hh@163.com (H.H.); conandoyle1887@163.com (Y.L.); zff959595@163.com (F.Z.); 2Department of Psychology, University of Chinese Academy of Sciences, Beijing 100101, China; 3CAS Key Laboratory of Behavioral Science, Institute of Psychology, Beijing 100101, China; 4Hubei Provincial Key Laboratory of Developmentally Originated Disease, Wuhan 430060, China

**Keywords:** schizophrenia, humans, social cognition, social interaction, theory of mind, parahippocampal gyrus, magnetic resonance imaging, fMRI, functional connectivity, psychophysiological interaction

## Abstract

Background: Theory of Mind (ToM) is an ability to infer the mental state of others, which plays an important role during social events. Previous studies have shown that ToM deficits exist frequently in schizophrenia, which may result from abnormal activity in brain regions related to sociality. However, the interactions between brain regions during ToM processing in schizophrenia are still unclear. Therefore, in this study, we investigated functional connectivity during ToM processing in patients with schizophrenia, using functional magnetic resonance imaging (fMRI). Methods: A total of 36 patients with schizophrenia and 33 healthy controls were recruited to complete a ToM task from the Human Connectome Project (HCP) during fMRI scanning. Psychophysiological interaction (PPI) analysis was applied to explore functional connectivity. Results: Patients with schizophrenia were less accurate than healthy controls in judging social stimuli from non-social stimuli (Z = 2.31, *p* = 0.021), and displayed increased activity in the right inferior frontal gyrus and increased functional connectivity between the bilateral middle temporal gyrus and the ipsilateral parahippocampal gyrus during ToM processing (AlphaSim corrected *p* < 0.05). Conclusions: Here, we showed that the brain regions related to sociality interact more with the parahippocampal gyrus in patients with schizophrenia during ToM processing, which may reflect a possible compensatory pathway of ToM deficits in schizophrenia. Our study provides a new idea for ToM deficits in schizophrenia, which could be helpful to better understand social cognition of schizophrenia.

## 1. Introduction

Schizophrenia is a common severe mental disease affecting more than twenty million people in the world [1]. The etiology of schizophrenia is still controversial. The dopamine hypothesis believes that schizophrenia is dopamine dysregulation in the context of a compromised brain [2]. According to the neurodevelopmental hypothesis, pathological changes in schizophrenia caused by genetics and the environment begin before the brain approaches its adult anatomical state in adolescence [3]. Common symptoms of schizophrenia include positive symptoms such as delusions and hallucinations, and negative symptoms such as social withdrawal and apathy. In addition, cognitive impairments are considered as a core feature of schizophrenia and exist in about 80% of patients [4]. The involved domains encompass a wide range of functions, including neurocognitive domains, such as attention, memory and executive function, as well as social cognitive domains, such as theory of mind (ToM) [5].

ToM (also called mental state attribution or mentalizing) typically involves the ability to infer the intentions, dispositions, and beliefs of others [6]. In addition to patients in an acute psychotic stage, decreased ToM abilities also exist in patients during remission, in relatives of schizophrenia patients, and in individuals who bear a risk of developing psychosis [7]. Social cognition, particularly ToM, which strongly associates with social functioning [8], is a predictor of functional outcome in schizophrenia [9].

Neuroimaging studies have suggested that there are two distinct large-scale neural networks involved in ToM [10]: One is the so-called “mentalizing network”, including the main cortical areas, such as the prefrontal cortex (PFC), the posterior–parietal junction (TPJ), the posterior parietal cortex (PPC), the superior temporal sulcus (STS), the temporal poles, and the cingulate cortex [11,12,13]. The other is the mirror neuron system, including the premotor cortex, the inferior frontal gyrus (IFG) and the inferior parietal lobule (IPL) [14], which are involved in action understanding and imitation [15]. The mentalizing network and the mirror neuron system play different and complementary roles in processing social intentions. The mirror neuron system is automatically activated when attending others’ actions and their goals when observing a moving body, and the mentalizing network is recruited when others’ intentions cannot be derived from available visuomotor cues and must be inferred in terms of internal mental states [16,17].

Decreased grey matter volume [18] and abnormal spontaneous activity [19] in rest within a wide range of brain regions is regarded as the neural basis of schizophrenia, and the structural as well as functional abnormalities of specific brain regions could be associated with specific symptoms of schizophrenia. Prefrontal cortex dysfunction is considered as the main pathophysiological mechanism of impaired cognitive control in schizophrenia [20], and decreased activation in the prefrontal cortex exists consistently in patients with schizophrenia during different cognitive paradigms [21]. In addition, reduced neural activity in the left precentral gyrus or Broca, as well as reduced right hemispheric hippocampal volume, is associated with impaired motor control and verbal memory, which are common domains of neurocognition [22].

With most research focusing on the neural basis of impaired neurocognition in schizophrenia, a few studies have explored the brain changes in schizophrenia during social cognition processing such as ToM. A meta-analysis showed that under-activation in schizophrenia patients was consistently found in the cortical regions, which are normally specialized for social cognition, such as the medial–prefrontal cortex (MPFC), the left orbitofrontal cortex, the right premotor cortex, and the left lateral occipito-temporal cortex (posterior temporo-parietal junction), and over-activation was found in the attention-related regions, such as the bilateral, dorsal section of the temporo-parietal junction [7]. Such over-activation may be associated with a compensatory response that recruits alternative strategies to foster ToM performance, and socio-cognitive deficits such as an impaired ToM might be explained by less specialized social brain processes for which the brain is not able to compensate. However, whether such compensation exists and how it works is still a controversial issue.

As for the relationship between functional connectivity and ToM deficits in schizophrenia, resting-state fMRI studies have found reduced within-network connectivity of the mentalizing network and the mirror network [10,23] and reduced connectivity between the mirror neuron system and other brain networks [23,24] in schizophrenia. In addition, the dysconnectivity of the default mode network (DMN), a network largely overlapping with the mentalizing network that is activated during rest [25], has also been proven to be associated with ToM deficits in patients with schizophrenia [26,27]. However, the results of resting-state fMRI cannot reflect the brain state during ToM processing directly, and thus, task-based fMRI studies are necessary. To date, only two fMRI studies have explored functional connectivity in schizophrenia using ToM tasks, obtaining inconsistent results. One of the studies found increased functional connectivity between the left inferior frontal gyrus and caudate nucleus [28], and the other found decreased functional connectivity between the bilateral posterior superior temporal sulcus [29].

To further clarify the changes of functional connectivity in schizphrenia during ToM processing, we used task-based fMRI and psychophysiological interaction (PPI) analysis in the present study. The ToM task we used is from the Human Connectome Project (HCP) [30], and previous studies have used similar tasks to explore ToM deficits in patients with schizophrenia [28,31,32,33,34]. Specifically, we will focus on: (1) brain activation patterns of two groups during the ToM task and the changes of activation in patients with schizophrenia, and (2) the changes of functional connectivity between the task-activated brain regions and the whole brain in patients with schizophrenia.

## 2. Materials and Methods

The whole process of this study was conducted as in the flow chart presented in Figure 1.

### 2.1. Participants

We recruited 40 schizophrenia patients from the inpatients and outpatients at the department of Psychiatry, Renmin Hospital of Wuhan University (Wuhan, China). All patients were assessed by a senior psychiatrist (H.L. Wang) according to the DSM-IV Disorder-Clinical Version (SCID-CV) [35]. We also recruited 34 age- and gender-matched healthy control participants. All participants were Chinese and right-handed. All patients were not associated with the symptom of impulsion and were able to complete continued assessment and MRI scanning. Symptom severity was assessed using the Positive and Negative Syndrome Scale (PANSS) [36]. A total score of PANSS greater than 70 was the necessary inclusion criteria. Twelve patients were not treated with any antipsychotics when they were recruited, and the dosages of antipsychotics of the rest of patients were converted to chlorpromazine-equivalent dosage. All participants completed PANSS assessment and MRI scanning within one week after recruited. Exclusion criteria for both groups were a lifetime history of serious physical diseases, neurological disorders, brain damage, substance abuse, and MRI-scan contraindications.

The Ethics Committee of Renmin Hospital of Wuhan University approved the study. Written informed consent was obtained from all participants after a complete description of the study.

### 2.2. Stimuli and Task Design

The present study used the social cognition paradigm from the Human Connectome Project (HCP) [30]. The paradigm has previously been used to investigate ToM deficits in schizophrenia [31,34]. In this task, two moving geometric shapes interacted as either a social (ToM) fashion resembling a social interaction among individuals or a non-social (random) fashion on a screen. After each animation, participants were required to judge which way the geometric shapes interacted from the following three options, including mental interaction, indefinitely, and irrelevant. “Mental interaction” is the correct option for ToM animation, and “irrelevant” is the correct option for random animation. Specifically, we prepared a total of five animations, of which two were for ToM interaction and three were for random interaction. Each animation lasted 20 s, with a fifteen-second segment presented with only a black cross on the screen (i.e., fixation period) between adjacent animations. Two different types of animations were presented alternately in a fixed order. After each animation, there was 3 s for participants to make a judgment. The flow chart of the task is presented in Figure 2.

Participants completed the task while undergoing an MRI scan, and the animations were presented using E-Prime 2.0 (Psychology Software Tools, Inc.; Sharpsburg, PA, USA) on a computer screen. Responses were made using an MR-compatible button-press.

### 2.3. MRI Data Acquisition

Structural and functional MRI data were acquired on a 3-T GE scanner (Discovery CT750, Willowick, OH, USA) at the Radiology Department of Remin Hospital of Wuhan University. Functional data were collected using echo planar imaging (EPI) utilizing the following parameters: TR = 2000 ms; TE = 30 ms; flip angle = 90°; matrix = 64 × 64; FOV = 220 mm. A total of 99 volumes (including 4 dummy scans) were obtained, and each volume consisted of 32 interleaved axial slices (slice thickness = 4 mm; gap = 0.6 mm). For anatomical reference, high-resolution structural imaging was also applied with the following parameters: TE = 3 ms; flip angle = 7°; matrix = 256 × 256; FOV = 256 mm; slice thickness = 1 mm; and no gap in 188 sagittal slices.

### 2.4. Data Analyses

#### 2.4.1. Demographics, Clinical, and Behavioral Data Analysis

All analyses were carried out in SPSS (version 26.0). Accuracy scores in two different types of tasks for each participant were calculated by counting the correct response number. Mann–Whitney U test were applied to analyze the differences in accuracy scores. Independent sample *t* tests were applied to analyze the differences in age and years of education. Chi-square test was applied to analyze the difference in gender. *p* < 0.05 is considered as a statistically significant difference.

#### 2.4.2. MRI Data Preprocessing

MRI data were analyzed using Statistical Parametric Mapping Version 12 (SPM12) (http://www.fil.ion.ucl.ac.uk/spm, accessed on 15 March 2023) under MATLAB R2013b. For each participant’s data, preprocessing was applied in the following steps: slice timing correction; head motion correction/realignment using rigid-body transform to generate movement parameters; segmentation of T1 images into grey matter, white matter, and cerebrospinal fluid; coregistration of mean fMRI images to T1 images; spatial normalization of fMRI images to standard MNI space by applying realignment and segmentation deformation, and fMRI images were resampled to 3 × 3 × 3 mm^3^
; at last, spatial smoothing using a 6 mm Gaussian kernel. According to the movement parameters, five participants including four patients and one healthy control were excluded for severe head motion (more than 3 mm maximum displacement and/or 3° of angular motion in x, y, or z directions). Therefore, there were 36 patients with schizophrenia and 33 healthy controls in the subsequent statistical analysis in total.

#### 2.4.3. GLM Analysis

Whole-brain statistical analyses were performed on SPM 12 using the general linear model (GLM) [37]. For first-level analysis, the time series of each participant’s images were high-pass filtered using a discrete cosine set (cutoff is 128 s), and serial correlations were accounted for using an autoregressive AR (1) model. Separate regressors (ToM, random, fixation) were modeled to convolve with a canonical HRF. The movement parameters were included as variables of no interest. For each participant, the contrasts of ToM > random and random > ToM were computed.

For second-level analysis, individual contrast images from the first level were entered into a random-effect analysis model. To determine the brain activation patterns of the two groups, one-sample t tests were performed for each group separately at the condition of ToM > random using a threshold of *p* < 0.05 corrected for multiple comparisons using false discovery rate (FDR) with a cluster extent threshold of 50 voxels. For between-group comparison, two-sample t tests were performed to identify differences in brain activation at the conditions of ToM > random and ToM < random using a threshold of *p* < 0.005 at the voxel level and *p* < 0.05 at the cluster level corrected with the AlphaSim correction. Age, gender, education, and the head motion parameters (framewise displacement) were included as covariates in the second level analysis.

#### 2.4.4. Psychophysiological Interaction Analysis

Psychophysiological interaction analyses (PPI) [38] were performed on SPM 12, which were used to measure the correlations of time series of the volume of interest (VOI) with other brain areas. The above activation analyses found that the left and the right middle temporal gyrus (MTG) showed the most significant positive activation in both groups and the right inferior frontal gyrus (rIFG) showed significant between-group differences in brain activation. Therefore, we chose these 3 regions as VOIs to implement PPI analyses separately. Specifically, for each participant, the time series of VOIs were extracted from the first-level analyses using a sphere of 6 mm radius, with the peak coordinates of the right MTG (x = 66, y = −48, z = 3), the left MTG (x = −60, y = −57, z = 0) and the right IFG (x = 57, y = 27, z = 12), at uncorrected threshold *p* < 0.05 with a minimum cluster size of 5 voxels.

Then, the subject-level GLM model for PPI analysis was built, which entailed three main regressors: the physiological variable, the psychological variable, and the PPI variable. The PPI variable was calculated as the element product of the deconvolved VOI time series (i.e., physiological variable) and a vector coding for the effect of the task (ToM > random, i.e., psychological variable). These three terms was susequentently re-convolved with the HRF. Then, the beta images of the interaction regressor from all participants were loaded into subsequent second-level PPI analysis to identify group differences using a two-sample t test with a threshold of *p* < 0.005 at the voxel level and *p* < 0.05 at the cluster level corrected with the AlphaSim correction. Similarly, age, gender, education, and head motion parameters were included as covariates. Functional connectivity between each VOI and other brain areas was analyzed separately.

#### 2.4.5. Correlation Analysis

For each participant, we extracted the beta value of functional connectivity between brain regions, which showed between-group differences at the condition of ToM > random using RESTplus v1.25 (http://restfmri.net/forum/, accessed on 15 March 2023). Then, Pearson correlation analyses were performed to evaluate the relationship to the PANSS scores, and the Spearman correlation analyses were performed to evaluate the relationship to the accuracy scores of ToM.

#### 2.4.6. Exploratory Analysis

*Treated vs. untreated*. A subgroup analysis was performed to compare treated (n = 24) and untreated (n = 12) patients in the above-mentioned brain activation and PPI analyses.

*Resting-state functional connectivity*. To explore whether the functional connectivity in the TOM task was also impaired during rest, we conducted an exploratory ROI-to-ROI resting-state functional connectivity analyses. The resting-state fMRI data were analyzed with SPM12 and Restplus running under MATLAB R2013b. The three VOIs and the brain regions with significant differences between the patients and the controls were selected as ROIs. Pearson’s correlation coefficients were calculated between the rs-fMRI time series of ROIs. Multiple comparisons were performed by the connection level false-discovery rate (FDR)-corrected *p* < 0.05.

## 3. Results

### 3.1. Demographic and Clinical Information

Demographic information of all the participants and the clinical information of the patients with schizophrenia are demonstrated in Table 1. There were no significant differences in gender and age between patients with schizophrenia and healthy controls (*p* > 0.05), but the healthy controls had longer years of education than the patients with schizophrenia (*p* < 0.001).

### 3.2. Behavioral Results of fMRI Paradigm

There were no significant differences between healthy controls and patients with schizophrenia at judging non-social motion (Z = 1.16, *p* = 0.246). However, healthy controls were more accurate in judging social motion than patients with schizophrenia (Z = 2.31, *p* = 0.021, Table 1). In total, the accuracy score of patients with schizophrenia was lower than healthy controls (Z = 2.17, *p* = 0.030).

### 3.3. Brain Activation

Brain activation patterns in the healthy controls and the patients of schizophrenia at the condition of ToM > random are presented in Figure 3A,B. In both groups, ToM conditions produced increased activity bilaterally in the middle temporal gyrus (MTG), the superior temporal gyrus (STG), the fusiform gyrus, and in the left inferior temporal gyrus (ITG) and the left inferior parietal lobule (IPL). The between-group result at the condition of ToM > random are presented in Figure 3C. Compared with the healthy controls, the patients with schizophrenia showed increased activity in the right inferior frontal gyrus (rIFG) (peak coordinates: x = 57, y = 27, z = 12; cluster size = 35; t = 3.07; *p* = 0.002). No brain regions showed group differences in activation at the condition of ToM < random.

### 3.4. Functional Connectivity: PPI

We selected three brain regions, including two regions showing the most significant positive activation in both groups in the above within-group results (the left and right middle temporal gyrus, MTG) and one region showing between-group differences in brain activation (the right inferior frontal gyrus) as seeds to implement PPI analyses. With the left MTG as a seed region, patients with schizophrenia displayed significantly increased functional connectivity to the left parahippocampal gyrus (PHG) and the left fusiform gyrus, compared with healthy controls at the condition of ToM > random. Similary, with the right MTG as a seed region, patients with schizophrenia displayed increased functional connectivity to the right PHG compared with healthy controls. With the right IFG as a seed region, no significant group differences in functional connectivity were found. The results are shown in Table 2 and Figure 4.

### 3.5. Correlation Analysis

A significant positive correlation was found between functional connectivity between the right MTG and the right PHG and the accuracy score of ToM (r = 0.431, *p* = 0.028) in the healthy controls but not in the schizophrenia patients. The functional connectivity between the left MTG and the left PHG showed no significant correlation to the accuracy score of ToM neither in the healthy controls nor in the schizophrenia patients. No significant correlation was found between these functional connectivity values and the total score or each subscale score of PANSS in the patients with schizophrenia.

### 3.6. Exploratory Analysis

In the subgroup analysis, there were no group differences in brain activation and PPI analyses between treated and untreated patients.

In ROI-to-ROI resting-state functional connectivity analyses, no significant correlations were found between the time series of ROIs, including the left MTG to the left PHG, and the right MTG to the right PHG.

### 3.7. Summarize

To sum up, the patients with schizophrenia in this study have more difficulty understanding social intentions, and the brain regions related to sociality show more cooperation with the parahippocampal gyrus in patients with schizophrenia during understanding of social intentions, and such abnormalities are not found at rest.

## 4. Discussion

In this study, we explored the neural basis of ToM deficits in patients with schizophrenia using a social cognitive fMRI paradigm from HCP, and healthy participants in this study showed similar activation patterns with the results of Barch et al. [30]. The main finding of this study is that patients with schizophrenia showed significantly worse performance on ToM but increased activity in the right IFG and increased functional connectivity between the MTG and the PHG compared with healthy controls. We assume that the increased functional connectivity observed in patients with schizophrenia may reflect a possible compensatory pathway of ToM deficits involving the parahippocampal gyrus.

The patients with schizophrenia were less accurate than the healthy controls in judging social stimuli from non-social stimuli, which implies the deficits in ToM, and was consistent with previous studies [39]. One of the explanations is that this relates to the production of erroneous interpretations of mental states [40]. The patients with schizophrenia displayed increased brain activation in the right IFG compared with the healthy controls. The abnormal activity in the IFG has been found in many previous studies using the ToM task in patients with schizophrenia [28,31,32,41,42,43,44]. IFG was widely recognized to be an important component of the mirror neuron system [45], which is involved in understanding and imitation. Earlier studies [31,46] using a similar paradigm, recruiting twenty-three male patients with schizophrenia and twenty-two healthy males, found significantly diminished activity in the bilateral IFG in male treated patients. However, in the subsequent studies that controlled confounding factors, including sex and treatment, the results tended to support that patients with schizophrenia showed increased activity in IFG during the ToM task [28,32], or in the anterior medial prefrontal cortex [33], as is partly in accordance with the present study. The ToM task in this study belongs to the implicit ToM paradigm [47]. In this study, the participants were asked to identify the animation type. Therefore, there could be task-related brain regions that can be actively activated. Because of the low efficiency of neural processing [48], patients with schizophrenia need to mobilize more brain region activation to complete the task of ToM compared with healthy controls, which could also explain the result that patients perform worse on the ToM task.

According to PPI analysis, we found that the patients with schizophrenia showed increased functional connectivity between the bilateral MTG and ipsilateral PHG. The MTG is closely related to language, emotion, and social cognition [49]. The left and right MTG were observed in this study as the most activated brain regions either in the healthy control group or in the schizophrenia group at the condition of ToM > random. This is consistent with a recent meta-analysis [50], which showed that the bilateral MTG were most significantly activated during ToM tasks in patients with schizophrenia. PHG is known to be involved in scene recognition, episodic memory, spital navigation, memory encoding and retrieval [51]. The processing of ToM also needs the involvement of memory [52]. For example, participants in this study may recall a similar social experience in the past when facing geometric shape movement to make a judgment. The deficits of ToM also exist in patients with Alzheimer’s disease [53], and one study attributed it to the impairment of episodic memory [54]. Increased functional connectivity to PHG in this study can be interpreted as more using episodic memory and memory retrieval, which could be regarded as a possible compensatory way.

In addition to memory, a few fMRI studies also discovered the role of functional connectivity between the MTG and the PHG in other functions. In the study of Antoine et al. [55], significant correlations were found between anosognosia in Alzheimer’s disease and decreased connectivity between the bilateral posterior parahippocampal cortex and the right middle temporal gyrus. Yu et al. [56] found that the right MTG showed significant functional connectivity with the left PHG in young adults with autistic-like traits while completing an emotion regulation task. Another study found that functional connectivity between the PHG and the MTG moderates the relationship between problematic mobile phone use and depressive symptoms [57]. According to the hypothetical neuropathways of ToM in schizophrenia given by Weng et al. [50], information of visual signals is continuously processed from the MTG to the right PHG during ToM tasks. A recent study defined a common fronto-temporo-parietal network involved in the comprehension of human events such as the theory of mind, including the MTG, the PHG and many other brain regions [58]. Therefore, the increased functional connectivity between the MTG and the PHG in patients with schizophrenia may reflect that the brain networks involved in ToM are more strongly activated during ToM processing.

The PHG is now identified as a node of the DMN [59]. A fMRI study found that patients with schizophrenia showed increased connectivity/weaker suppression of the DMN during a ToM task compared with healthy controls [60]. The increased activity of DMN is often explained as compensatory effort for brain function [61]. There is a possible explanation that the increased DMN connectivity could then act as a compensatory mechanism to upregulate the activity in the MTG, which results in no difference in localized MTG activity between patients with schizophrenia and healthy controls in this study. Subsequently, we performed an exploratory analysis using the same sample to verify whether such hyperconnectivity between MTG and PHG also existed in patients during rest using resting-state fMRI. It showed that there is no significant difference between the two groups in functional connectivity during rest between the MTG and the PHG. Thus, we propose that such hyperconnectivity between the MTG and the PHG in patients with schizophrenia may reflect a specific brain change depending on ToM deficit.

However, it seems that such compensation is unsuccessful, because patients with schizophrenia still have significantly worse ToM performance, and correlation analyses showed that only in healthy controls rather than in patients with schizophrenia was the functional connectivity between the right MTG and the right PHG positively correlated with ToM performance.

Previous studies have suggested the effects of antipsychotic drugs to brain function, primarily in the frontal and striatal regions [62,63], and Sambataro et al. [64] found that treatment with olanzapine was associated with increases in DMN connectivity. However, there was no group differences in whole-brain activation and functional connectivity between the MTG and the PHG between treated and untreated patients in the subgroup analysis. This supported that the changes in brain region function in schizophrenia involving ToM deficits were more likely to result from illness instead of from treatment.

Social cognition, particularly ToM, is strongly associated with social functioning. Our study provides a theoretical basis for the target of therapy to improve social cognition in schizophrenia and the evaluation of therapeutic effects. The ultimate goal for us is to find the pathological mechanism of impaired social cognition in schizophrenia so that effective treatment could be developed to improve social cognition ability of schizophrenia patients and to solve the question of recession. However, there is still much challenge, and future research could consider a larger sample size, multiple ToM measurements and multimodel MRI to explore the neural basis of ToM deficits in schizophrenia in depth.

A few limitations need to be considered when analyzing these results. First, we simplified the paradigm on the number of animations to ensure that all participants could complete it seriously. Therefore, the degree of brain activation of task-related regions might not be as good as previous studies due to less task load, which may affect the reliability of the subsequent analysis. Second, one third of patients in this study were treated with antipsychotic drugs. Thus, we performed a subgroup analysis to compare brain function between treated and untreated patients, which showed no significant differences between the two groups in brain activation and functional connectivity analyses. Finally, the single index in evaluating behavioral performance could help to reflect all respects of the ability of ToM.

## 5. Conclusions

In summary, this study has found that patients with schizophrenia displayed increased brain activation in the rIFG and increased functional connectivity between the MTG and the PHG during ToM processing. We propose that such hyperconnectivity between the MTG and the PHG existing in patients with schizophrenia during ToM processing specifically may reflect a possible compensatory mechanism against ToM deficits. Our study provides a new idea for ToM deficits in schizophrenia involving the parahippocampal gyrus, which could provide reference for further research to explore the contribution of the parahippocampal gyrus to schizophrenia at the molecular level. Functional connectivity reflects the similarity of neural activity between different brain regions, instead of anatomical connections. Therefore, future studies could integrate task-related fMRI with other modal imaging measures such as diffusion tensor imaging to further explore the anatomical changes behind the functional connectivity changes. 

## Figures and Tables

**Figure 1 brainsci-13-00692-f001:**
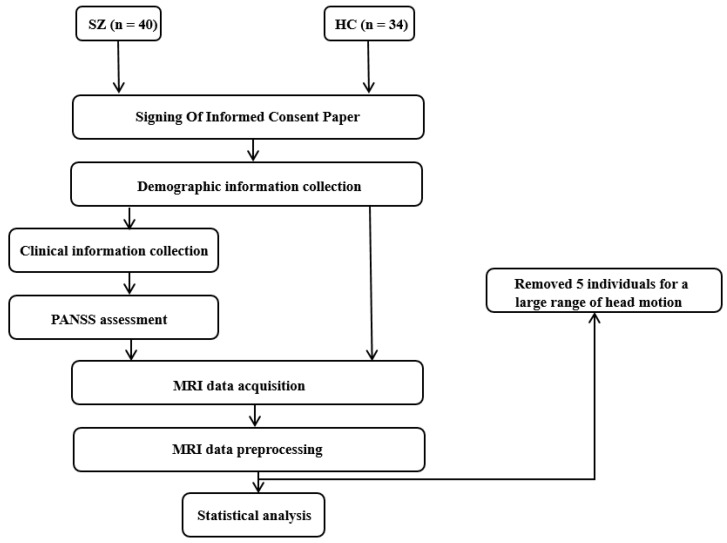
Flow chart of the whole study.

**Figure 2 brainsci-13-00692-f002:**
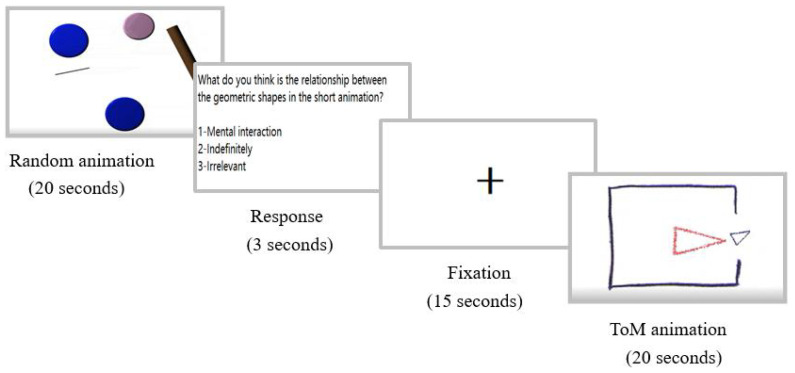
Flow chart of the animated triangle task.

**Figure 3 brainsci-13-00692-f003:**
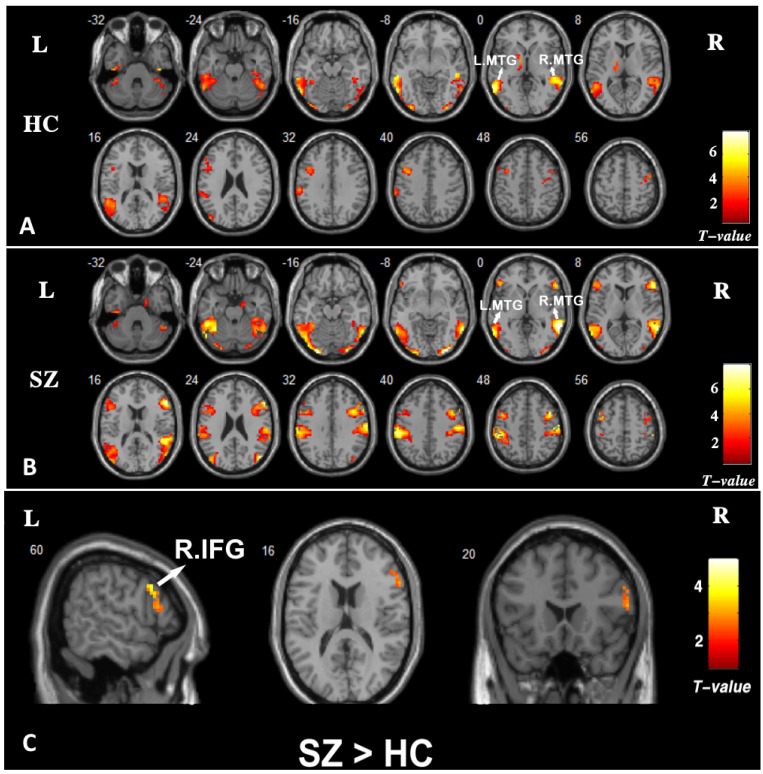
Brain activation patterns in the groups of (**A**) healthy controls (HC) and (**B**) schizophrenia (SZ) at the condition of ToM > random. Threshold at *p* < 0.05, FDR-corrected for multiple comparisons and cluster extent > 50 voxels. (**C**) Between-group brain activation results at the condition of ToM > random. Threshold at *p* < 0.005 at the voxel level and *p* < 0.05 at the cluster level corrected with the AlphaSim correction. Color bars represent T values. MTG, middle temporal gyrus. IFG, inferior frontal gyrus.

**Figure 4 brainsci-13-00692-f004:**
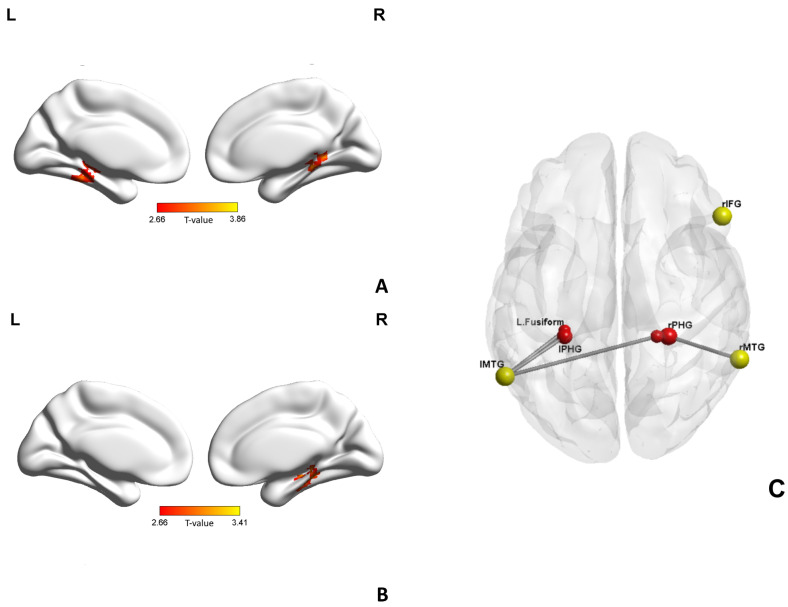
Results of psychophysiological interaction (PPI) analysis. (**A**) With the left middle temporal gyrus (MTG) as a seed, the patients with schizophrenia displayed increased functional connectivity between the left MTG and the left fusiform gyrus and the left parahippocampal gyrus (PHG). (**B**) With the right middle temporal gyrus (MTG) as a seed, the patients with schizophrenia displayed increased functional connectivity between the right MTG and the right PHG. (**C**) Overall PPI results. Seed regions are indicated as yellow spheres.

**Table 1 brainsci-13-00692-t001:** Demographic, clinical, and behavioral characteristics of all participants.

	HC (n = 33)	SZ (n = 36)	T/$\chi^{2}$/Z	*p* Value
Age (Years)	26.45 ± 4.50	24.44 ± 3.96	1.98	0.052
Male/Female	14/19	18/18	0.40	0.529 ^a^
Education (Years)	15.70 ± 2.02	12.94 ± 2.50	5.00	<0.001 *
Age of onset (Years)		22.19 ± 4.13		
Duration of illness (Years)		2.22 ± 2.36		
CPZ equivalent (mg)		455.42 ± 236.42		
PANSS				
positive		18.11 ± 4.09		
negative		18.34 ± 5.05		
general		40.51 ± 4.97		
total		78.25 ± 7.03		
Accuracy scores of tasks				
Random	1.73 ± 0.91	1.47 ± 1.03	1.16	0.246 ^b^
ToM	1.70 ± 0.53	1.39 ± 0.60	2.31	0.021 ^b,^*
Total	3.42 ± 1.12	2.86 ± 1.13	2.17	0.030 ^b,^*

CPZ equivalent is calculated from treated patients (n = 24). HC, healthy controls; SZ, patients with schizophrenia; CPZ, chlorpromazine; PANSS, Positive and Negative Syndrome Scale; ^a^ Chi-square test; ^b^ Mann–Whitney U test; * significant differences (*p* < 0.05).

**Table 2 brainsci-13-00692-t002:** Psychophysiological interaction analysis at the condition of ToM > random (AlphaSim corrected *p* < 0.05).

Conditions	Seed	Structures	Cluster Size	Peak Coordinates x,y,z (mm)	*p* Value
SZ > HC	L.MTG	L.Parahippocampal gyrus	54	−30 −36 −6	3.43
		L.Fusiform gyrus	26	−33 −42 −18	2.92
		R.Parahippocampal gyrus	30	21 −36 −9	3.37
	R.MTG	R.Parahippocampal gyrus	79	27 −36 −12	3.25

SZ, patients with schizophrenia; HC, healthy controls; MTG, the middle temporal gyrus.

## Data Availability

The data can be made available upon reasonable request to the corresponding author. https://doi.org/10.6084/m9.figshare.22086398.v1, accessed on 15 March 2023.

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
