# Peer review of "Increased Functional Connectivity Involving the Parahippocampal Gyrus in Patients with Schizophrenia during Theory of Mind Processing: A Psychophysiological Interaction Study"

_brainsci, 2023, doi:10.3390/brainsci13040692_

Round 1
Reviewer 1 Report
1: In the materials and methods 2.1 participant section, if a graphical representation of the whole recruitment process if provided would be too easy to understand for the readers.
2. Can the authors provide detailed in inclusion, exclusion criteria, primary and secondary outcome measures in a tabular form along with participants who were already on antipsychotics and what type of antipsychotics were they on at the time of participation?
Author Response
Dear Professor,
Thank you for valuable comments and suggestions. And there is an attachment including our reply to your comments and suggestions.
We are looking forward to your further review.
Kind regards,
Xucong Qin

Reviewer 2 Report
4 April 2023
Manuscript ID: brainsci-2320460
Type: Article
Title: ‘Increased Functional Connectivity Involving the Parahippocampal Gyrus in Patients with Schizophrenia during Theory of Mind Processing: A PPI Study’ by Qin X et al., submitted to Brain Sciences
Dear Authors,
The present review article by Qin X and colleagues, entitled ‘Increased Functional Connectivity Involving the Parahippocampal Gyrus in Patients with Schizophrenia during Theory of Mind Processing: A PPI Study’ is a well-written and useful manuscript on the schizophrenia and theory of mind processing.
The main strength of this manuscript is that it addresses an interesting and timely question, describing that patient with schizophrenia had increased functional connectivity involving the parahippocampal gyrus during theory of mind processing.
In general, I think the idea of this manuscript is really interesting and the authors’ fascinating observations on this timely topic may be of interest to the readers of Brain Sciences. However, some comments, as well as some crucial evidence that should be included to support the author’s argumentation, needed to be addressed to improve the quality of the manuscript, its adequacy, and its readability prior to the publication in the present form, in particular reshaping parts of the Introduction and Discussion sections by adding more evidence and theoretical constructs.
Please consider the following comments:
1. Title: Please avoid using abbreviation in the title.
2. A graphical abstract that will visually summarize the main findings of the manuscript is highly recommended.
3. Abstract: I would like the authors may focus on proportionally presenting the background, methods, results, and conclusion. The background should include the general background (one to two sentences), the specific background (two to three sentences), and current issue addressed to this study (one sentence), leading to the objectives. In this subsection I would like the authors to lay out basic information, problem statement, and the authors’ motivation to break off. The methods should clarify the authors’ approach such as study design and variables to solve the problem and/or to make progress on the problem. The results section must state the results in numbers and clarify their statistical significance. Are the results statistically significant? This subsection should close with a paragraph which puts the results into a more general context. The conclusion should include one sentence describing the main result using such words like “Here we show”. The conclusion should write the potential and the advance this study has provided in the field and finally a broader perspective (two to three sentences) readily comprehensible to a scientist in any discipline.
4. Keywords: Please list ten keywords from Medical Subject Headings (MeSH) (https://meshb.nlm.nih.gov/) and use as many as possible in the title and in the first two sentences of the abstract.
5. In general, I would like the authors to clarify the following points: a)
What is the parahippocampal gyrus and how does it relate to theory of mind processing?
b) How was the study conducted and what were the results?
c) What implications does this study have for the treatment of schizophrenia?
6. Introduction: The ‘Introduction’ section is well-written and nicely presented, with a good balance of descriptive text and information. I recommend that the authors focus on presenting the following elements of introduction with several paragraphs consisting of up to 1000 words, to introduce the main constructs of this study, which should be understood to a reader in any discipline and make persuasive enough to put forward “the main purpose of current research the authors has conducted and the specific purpose the authors has intended by this study. I would like to encourage the authors to present the introduction starting with the general background, proceeding to the specific background, and finally the current issue addressed to this study, leading to the objectives. Those main structures should be organized in a logical and cohesive manner. Furthermore, I believe that adding more information about pathophysiology and core features of schizophrenia will provide a better and more accurate background, because as it stands, this information is not highlighted in the text. In this regard, I would suggest adding more information on pathological neural substrates of neurodegeneration in schizophrenia, specifically on structural as well as functional abnormalities of specific brain regions (i.e., hippocampus and prefrontal cortex), and on related and on related effects on patients’ cognitive impairments. In my opinion, the authors could further explore significant structural brain alterations and impaired brain circuits in AD (https://doi.org/10.1016/j.tins.2022.04.003; https://doi.org/10.1111/psyp.14122) and focus on relationship between the molecular regulation of higher-order neural circuits and neuropathological alterations in this neurodegenerative disorder (https://doi.org/10.3390/cells11162607; https://doi.org/10.3390/biomedicines9050517).
7. Methods: I recommend opening this section with a short introductory paragraph regarding the study design and methodology. Also, I suggest citing more references to ensure the reliability and the integrity of evidence in the study design the authors have built and the methodology the authors applied to this study.
8. Results: I recommend that the authors close this section with a paragraph which put the results into a more general description.
9. Discussion: The authors need to present the independent discussion section with up to 1500 words and to focus on the following essential elements for discussion. Starting with an introductory paragraph, I would like the authors to present the summary of the previous section and to develop discussion on the potential of this study complementing as the extension of the previous work, the implication of the findings of this study, how this study could facilitate future research, the ultimate goal, the challenge, the knowledge and the technology necessary to achieve this goal, the statement about this field in general, and finally the importance of this line of research.
10. I would like the authors to make this section with a paragraph which would benefit from some thoughtful as well as in-depth considerations by the author as an expert to convey the take-home message, explaining the theoretical implication as well as the translational application of their research. I believe that it would be necessary to discuss theoretical and methodological avenues in need of refinement, as well as suggestions of a path forward understanding functional connectivity in schizophrenia.
11. References: Please follow the Journal’s guidelines. Page numbers should end with a period.
The manuscript contains three figures, two tables, and 55 references. I believe that the manuscript carries important value in studying theory of mind processing in patients with schizophrenia. I hope that, after these careful revisions, this paper can meet the Journal’s high standards for publication. I am available for a new round of revision of this paper.
I declare no conflict of interest regarding this manuscript.
Best regards,
Reviewer
Author Response
Dear Professor,
Thank you for valuable and helpful comments and suggestions. And we reply to your comments and suggestions in the attachment.
We are looking forward to your further review.
Kind regards,
Xucong Qin
E-mail: psy_qxc@163.com

Round 2
Reviewer 2 Report
10 April 2023
Manuscript ID: brainsci-2320460
Type: Article
Title: ‘Increased Functional Connectivity Involving the Parahippocampal Gyrus in Patients with Schizophrenia during Theory of Mind Processing: A PPI Study’ by Qin X et al., submitted to Brain Sciences
Dear Authors,
I am pleased to see that the authors took my comments seriously and have solved many issues I raised in the previous round of peer-review session. Currently, the manuscript is a well written and nicely presented research article reporting that patient with schizophrenia had increased functional connectivity involving the parahippocampal gyrus during theory of mind processing. Nevertheless, there remain a couple of crucial points the authors are expected to make their effort to revise prior to publication.
Comments:
1. Abstract: Please abridge the abstract to 200 words according to the journal’s guidelines (https://www.mdpi.com/journal/brainsci/instructions) without subheadings. I personally accept 200-220 words, but never more that 250 words. I leave here my previous suggestions, so that the authors maintain basic structures for a revised manuscript. I would like the authors may focus on proportionally presenting the background, methods, results, and conclusion. The background should include the general background (one to two sentences), the specific background (two to three sentences), and current issue addressed to this study (one sentence), leading to the objectives. In this subsection I would like the authors to lay out basic information, problem statement, and the authors’ motivation to break off. The methods should clarify the authors’ approach such as study design and variables to solve the problem and/or to make progress on the problem. The results section must state the results in numbers and clarify their statistical significance. Are the results statistically significant? This subsection should close with a paragraph which puts the results into a more general context. The conclusion should include one sentence describing the main result using such words like “Here we show”. The conclusion should write the potential and the advance this study has provided in the field and finally a broader perspective (two to three sentences) readily comprehensible to a scientist in any discipline.
2. The authors have clarified my questions. However, I would like the authors to use them in the abstract and the body of manuscript. I leave here my questions. a)
What is the parahippocampal gyrus and how does it relate to theory of mind processing?
b) How was the study conducted and what were the results?
c) What implications does this study have for the treatment of schizophrenia?
3. Introduction: This section has substantially improved. I would suggest to provide a general overview of pathogenesis and biochemical hallmarks of schizophrenia disorder, and add more information on neural substrates of schizophrenia, specifically on frontal lobe dysfunction, and on related effects on patients’ memory and learning impairments: this may provide a better understanding of prefrontal cortex’s key role and how its disrupted function may contribute to irregular behavioral responses (https://doi.org/10.3390/ijms24065926) and therefore to the development of many mood psychiatry disorders, including depression or anxiety, and those that are common in schizophrenia (DOI: 10.3390/biomedicines10122999).
4. Conclusion: I would like the authors to elaborate this section. I would like the authors to make this section with a paragraph which would benefit from some thoughtful as well as in-depth considerations by the author as an expert to convey the take-home message, explaining the theoretical implication as well as the translational application of their research. I believe that it would be necessary to discuss theoretical and methodological avenues in need of refinement, as well as suggestions of a path forward understanding functional connectivity in schizophrenia.
5. References: Please follow the Journal’s guidelines. Please abbreviate journal names with punctuaction.
The manuscript contains three figures, two tables, and 60 references. I believe that the manuscript carries important value in studying theory of mind processing in patients with schizophrenia. I hope that, after these careful revisions, this paper can meet the Journal’s high standards for publication.
I declare no conflict of interest regarding this manuscript.
Best regards,
Reviewer
Author Response
Dear Professor,
Thank you for valuable and helpful comments and suggestions. And we reply to your comments and suggestions in the attachment.
We are looking forward to your further review.
Kind regards,
